# The association between educational level and multimorbidity among adults in Southeast Asia: A systematic review

**Xiyu Feng, Matthew Kelly ⓘ\*, Haribondhu Sarma**

Department of Global Health, Research School of Population Health, The Australian National University, Canberra, Australia

\* matthew.kelly@anu.edu.au

## Abstract

**Data Availability Statement:** As this is a systematic review, data are available in the published articles included in the review.

**Funding:** The author(s) received no specific funding for this work.

### Background

In Southeast Asia, the prevalence of multimorbidity is gradually increasing. This paper aimed to investigate the association between educational level and multimorbidity among over 15-years old adults in Southeast Asia.

### Methods

We conducted a systematic review of published observational studies. Studies were selected according to eligibility criteria of addressing definition and prevalence of multimorbidity and associations between level of education and multimorbidity in Southeast Asia. The Newcastle-Ottawa Scale (NOS) was used to measure the quality and risk of bias. The methodology has been published in PROSPERO with registered number ID: CRD42021259311.

### Results

Eighteen studies were included in the data synthesis. The results are presented using narrative synthesis due to the heterogeneity of differences in exposures, outcomes, and methodology. The prevalence of multimorbidity ranged from 1.7% to 72.6% among over 18 years-old adults and from 1.5% to 51.5% among older people ($\geq$ 60 years). There were three association patterns linking between multimorbidity and education in these studies: (1) higher education reducing odds of multimorbidity, (2) higher education increasing odds of multimorbidity and (3) education having no association with multimorbidity. The association between educational attainment and multimorbidity also varies widely across countries. In Singapore, three cross-sectional studies showed that education had no association with multimorbidity among adults. However, in Indonesia, four cross-sectional studies found higher educated persons to have higher odds of multimorbidity among over 40-years-old persons.

**Competing interests:** The authors declare they have no competing interests

**Abbreviations:** NOS, Newcastle-Ottawa Scale; LMICs, Low- and middle-income countries; SES, Socioeconomic status; PRISMA, Preferred Reporting Items for Systematic reviews and Meta-Analyses; ASEAN, the Association of Southeast Asian Nations; CI, Cumulative incidence; IR, Incidence Rate; CI, Confidence interval; aOR, Adjusted odd ratio; SMHS, Singapore Mental Health Study; TB, Tuberculosis; WiSE, Well-being of the Singapore Elderly; IFLS, Indonesian Family Life Survey.

## Conclusions

Published studies have shown inconsistent associations between education and multimorbidity because of different national contexts and the lack of relevant research in the region concerned. Enhancing objective data collection such as physical examinations would be necessary for studies of the connection between multimorbidity and education. It can be hypothesised that more empirical research would reveal that a sound educational system can help people prevent multimorbidity.

## Introduction

Multimorbidity, also known as multiple chronic conditions, is generally defined as the presence of two or more chronic health conditions in a person at the same time [1]. Multimorbidity is more common in older adults (more than 60 years old), such as suffering from hypertension, diabetes and chronic kidney disease concurrently [2,3]. Due to global aging, multimorbidity is also becoming a global public health issue [2]. The magnitude of multimorbidity in low- and middle-income countries (LMICs) was estimated at 10% to 11% recently, and it is predicted to increase in the coming years [4]. Due to the complexity of symptoms and higher mortality rates, the treatment requirements for multimorbidity are more complicated than those for single diseases, and patients with multimorbidity often do not receive cost-effective treatment [5]. This situation may increase the economic and medical burden of these patients and lead to a reduced quality of life and greater damage to their physical and mental health [5,6].

In Southeast Asia, the prevalence of multimorbidity is also gradually increasing, from a prevalence of about 4.5% at the beginning of the 21st century to about 10% in recent years [4,7]. There are many factors contributing to the rising prevalence of multimorbidity in Southeast Asia, among which is the rapid socio-economic development of region and the concurrent growth in socioeconomic inequality [8–10]. Socio-economic development is connected to epidemiological transition and the growth of the prevalence of chronic diseases [9]. Moreover, inequalities in socioeconomic status (SES) may be reflected in unequal access to health care, participation in health activities, and life stressors, which would contribute to an increased burden of multiple chronic conditions [9,10]. Furthermore, lower socioeconomic groups who suffer from multimorbidity will suffer more because they have limited access to diagnosis and the burden of expensive treatment [8–10].

Education level generally refers to the highest attained level of education by individuals and is often classified into these levels: no education; elementary school; middle school, junior high school, senior high school, university or higher [9,10]. Moreover, education level, which is a key indicator of SES, may have a greater impact on the prevalence of chronic diseases compared with the other two factors (income and occupation) of SES. This is because in health studies on multimorbidity, measures of SES include income, occupation, and education, but education usually logically determines subsequent occupational and income development [9,10]. Moreover, education may also influence health literacy, leading to a potential role in reducing the prevalence of multimorbidity. Furthermore, educational inequalities are evident in Southeast Asian and are possibly influencing patterns of multimorbidity. In Southeast Asian nations, disparity in access to educational resources is common, often also along gender lines [11–13]. Thus, in this article, education level will be used as an exposure factor to measure its effect on multimorbidity in Southeast Asia.

Studies to date in Southeast Asia which have assessed associations between the education level and multimorbidity have had mixed outcomes. For example, some studies have concluded that lower education levels cause an increase in the prevalence of multimorbidity [14,15]. But other studies have established that the level of education was not connected with the prevalence of multimorbidity [16]. It is hypothesis that in high-income countries, education may prevent the occurrence of multimorbidity, but in LMICs, education may be a risk factor for multimorbidity [8,9]. Nevertheless, to date, no article has conducted a comprehensive systematic review of the association between multimorbidity and education level among populations living in Southeast Asia.

Therefore, this paper is the first attempt to provide an overview of studies on multimorbidity and educational attainment in Southeast Asia and to systematically evaluate published observational studies. The aim of this paper was to better understand the association between educational attainment and multimorbidity in Southeast Asia, which may help to identify potential causes of multimorbidity in these places and to design appropriate interventions to prevent or reduce the occurrence of multimorbidity.

## Methods

A systematic review of published articles reporting multimorbidity and educational level among adults in Southeast Asia was conducted using the terms of the Preferred Reporting Items for Systematic reviews and Meta-Analyses (PRISMA) statement (S3 Table) [17]. The methodology has been published in PROSPERO with registered number ID: CRD42021259311 (S1 File).

### Inclusion and exclusion criteria

Articles were selected based on the following inclusion criteria, (1) quantitatively designed observational studies; (2) studies reporting multimorbidity being defined as a person having two or more chronic conditions at the same time; (3) studies reporting or having available detailed data on associations between level of education and multimorbidity; (4) studies in which the study sites including either individual Southeast Asian countries or the Southeast Asian region, with the definition according to countries belonging to the Association of Southeast Asian Nations (ASEAN), which included Brunei, Cambodia, East Timor, Indonesia, Laos, Malaysia, Myanmar, Philippines, Singapore, Thailand and Vietnam; (5) the associations between education level and multimorbidity in the Southeast Asian region or Southeast Asian countries being reported in studies examining the regional level, such as at the level of LMICs or at the global level; (6) participants in studies being over 15 years of age; and (7) English-language studies being published between 1990 and 2021 due to the term of "multimorbidity" first being coined in the early 1990s [18].

The exclusion criteria were [18,19]: (1) book series and conferences; (2) qualitative studies; (3) non-observational studies; (4) studies reporting co-morbidity (studies with an index disease), such as multimorbidity among patients with diabetes, HIV or hypertension; (5) studies where detailed data was not available regarding the association between educational level and multimorbidity; (6) study sites not in Southeast Asian region or in Southeast Asian countries; (7) studies of examining the regional level such as LIMCs level and global level not reporting the associations between education level and multimorbidity in Southeast Asia region or Southeast Asian countries; (8) participants in studies was younger than 15 years; and (9) studies not published in English, and not published between 1990 and 2021.

## Search strategy and the selection of literature

The databases of Scopus, PubMed and ProQuest were used to search for relevant articles. We classified the search terms according to exposure, outcomes and location (S1 Table) [18,19]: (1) Exposure: 'education, literacy, educational status, educational level, educational attainment'. (2) Outcome: 'multimorbidity, multimorbidity, multimorbid, multiple morbidities, multiple morbidity, multiple conditions, multiple diseases, multiple chronic diseases, multiple chronic conditions, multiple illnesses, multiple diagnoses, multi-pathology'. (3) Location: 'Southeast Asia, Association of Southeast Asian Nation, ASEAN, Brunei, Cambodia, East Timor, Indonesia, Laos, Malaysia, Myanmar, Philippines, Singapore, Thailand, Vietnam". The 'AND' was used to the combination of search terms across the categories and 'OR' was combined within the categories. In addition, a term similar to the definition of multimorbidity is comorbidity, but in 2018, a distinction has been made between the definition of comorbidity and multimorbidity, and while both terms emphasize the co-existence of multiple chronic conditions in the same individual, the term of "comorbidity" means the combined effects of additional conditions with reference to the index chronic condition such as the comorbidity of diabetes, stroke, or depression [1]. Namely, comorbidity is the presence of one or more additional diseases, as a result of the presence of the index condition [19]. Although, these two terms have different definition, multimorbidity and comorbidity are commonly used interchangeably [1,8]. Thus, after the initial search, the addition of 'comorbidity' was added into our search to test if any articles had been missed through the exclusion of the term of 'comorbidity' and linguistic changes of the term of 'comorbidity' [18].

Furthermore, the studies in this paper only involved human participants, were published from 1990/01/01 to 2021/06/15, and had abstracts available. We used the hand search in the references of retrieved studies to identify additional relevant papers.

The first reviewer (XF) performed an initial screening of titles and abstracts for all keywords searched. The second reviewer (MK) conducted a 20% random sample of all references to ensure that eligible studies were not omitted [18]. Studies that met all of the above eligibility criteria were retained for full-text screening. Full-text screening was done independently by two reviewers (XF and MK). When disagreements arose, XF and MK resolved them through discussion. When agreement could not be reached, a third reviewer (HS) was consulted. Disagreements were eventually resolved by consensus.

## The assessment of quality

Two authors (XF and MK) independently assessed the risk of bias and study quality in cohort studies and case-control studies using the Newcastle Ottawa Scale (NOS), one of the risk of bias assessment tools recommended by the Cochrane Collaboration for use in observational studies [20], and cross-sectional studies using adaptations of the NOS (S2 File and S2 Table) [19,21]. For observational studies, the checklist focused on three aspects (Selection, Comparability and Exposure/Outcome).

The examined selection of observational studies according to NOS was shown in the table below (Table 1) [19,21]. Observational studies included cross-sectional studies, cohort studies and case-control studies. The selection table was based on a summary of the scored items for each of the observational studies in NOS (S2 File), with reference to other systematic reviews of multimorbidity that have applied NOS as an assessment tool to evaluate and examine the literature [19,21].

There were three levels of quality to assess the scores of the individual study in the table below (Table 2) [19,21]. The scores of NOS corresponding to the three levels mentioned in the

**Table 1. The examined selection of observational studies according to NOS.**

| | Cross-sectional studies (Seven items, maximum ten points) | Cohort studies (Eight items, maximum nine points) | Case-control studies (Eight items, maximum nine points) |
|---|---|---|---|
| **Selection** | (Up to five points) | (Up to four points) | (Up to four points) |
| | Representativeness; | Representativeness; | Adequate definition; |
| | Sample size; | Non exposed cohort; | Representativeness; |
| | Non-respondents; | Ascertainment; | Selection of controls; |
| | Ascertainment. | Demonstration. | Definition of controls. |
| **Comparability** | (Up to two points) | (Up to two points) | (Up to two points) |
| | On the basis of the study design or analysis and the control of confounders. | On the basis of the design or analysis. | On the basis of the design or analysis. |
| **Outcome/ Exposure** | (Up to three points) | (Up to three points) | (Up to three points) |
| | Assessment; | Assessment; | Ascertainment; |
| | Statistical test. | Enough long follow-up; | Same method of ascertainment; |
| | | Adequacy. | Non-response rate. |

table were summarized from similar levels and scores that appeared in other systematic reviews of multimorbidity [19,21].

Two reviewers (XF and MK) independently decided on the summary measurement of the relevant articles. The risk of bias was assessed as the sum of the scores for each item. Each reviewer independently determined an overall quality score for each article. It should be noticed that if one of the three aspects (Selection, Comparability and Exposure/Outcome) was given a zero score, the level was Poor, regardless of the total scores of all three aspects. The final article selection was based on the scores of the three aspects (Selection, Comparability and Exposure/Outcome). To be retained in our systematic review, articles should have had a quality score of four scores or above (S2 Table) [19,21].

## Data extraction

For each included study, we extracted the following information [18,19]: (1) the author(s) and publication year; (2) study country/location; (3) study design and study population (4) sample size: total number; (5) sample characteristics: the percentage of male (%); (6) sample characteristics: mean age and/or the range of age (years); (7) data collection; (8) the definition of multimorbidity and the number of conditions or diseases; (9) the subdivision of educational attainment; (10) the prevalence/incidence of multimorbidity (11) the prevalence/incidence of multimorbidity in terms of educational level; and (12) the main results of the association between multimorbidity and education (including analytical adjustments made).

## Data synthesis

Because of the different exposures, outcomes, and study methods, the included studies were judged to be heterogeneous in distinct ways such as populations, the definition of variables

**Table 2. Three levels of quality to assess the scores of the selected studies.**

| | Selection | Comparability | Outcome/Exposure | Total |
|---|---|---|---|---|
| **Good** | Four points or above | One point or above | Two points or above | Seven points and above |
| **Fair** | Two to three points | One point | One to two points | Five to six points |
| **Poor** | Zero to one point | Zero point | Zero point | Zero to four points |

and different confounders. Taking this heterogeneity into account, we were unable to perform a meta-analysis of the findings [18,22].

The findings regarding methods, exposures, and outcomes were described using narrative synthesis [18,22]. The prevalence of multimorbidity was extracted from individual studies. The extraction of the prevalence of multimorbidity was divided into three cases in these studies. (1) The prevalence of multimorbidity was reported directly in the study, thus we could extract data on the prevalence of multimorbidity directly from the article. (2) If the prevalence of multimorbidity was not directly reported in the study but it reported the total number of participants and the number of people with multimorbidity, we calculated this indicator using the number of participants as the denominator and the number of people with multimorbidity as the numerator. (3) If studies did not report the prevalence of multimorbidity, the total number of participants and the number of multimorbidity cases, we categorised those studies "not available" (N/A).

The available data regarding education level in the survey area were pooled in the same way to extract and calculate the prevalence (%) of multimorbidity at different education attainments. The prevalence of multimorbidity (%) at different education levels in the study was all obtained by calculation in selected studies. The prevalence of multimorbidity for a particular education level group was calculated by dividing the number of participants experiencing multimorbidity at that educational attainment level by the total sample number of participants. For studies that we could not calculate the prevalence of multimorbidity (%) at different educational levels because of lack of reporting the number of patients with multimorbidity (numerator) at different educational levels, we would put "N/A" to show the prevalence.

All studies reported the ratios of multimorbidity at different educational levels. Therefore, the ratios in this paper were used directly from the ratios given in each study.

## Results

### Yield of search strategy

The electronic and manual searches yielded 7,558 articles. After eliminating duplicate articles, 6,786 articles were selected for the title and abstract screening. After carefully screening the titles and abstracts, 36 articles were found to be available for full-text review. A further 18 articles were then removed due to their addressing comorbidity (17 articles) and being unable to obtain detailed data of multimorbidity in terms of educational level in Southeast Asia (1 article). Finally, 18 articles were included in this systematic review (Fig 1) [14–16,23–37]. Seventeen papers were cross-sectional studies [14–16,23–28,30–37], and one paper was a longitudinal study [29]. No studies were case-control studies.

### Study selection and characteristics

Table 3 detailed the key characteristics of the included studies. Two multi-country studies implemented in different Southeast Asian countries were included in this review [14,16]. Five studies were from Singapore [23,27,34–36], four studies were from Indonesia [24,28,30,31], three studies were from Vietnam (all from different parts of Vietnam) [15,26,32], two studies were from Thailand [33,37], one was from Myanmar [25] and one was from Malaysia [29]. Sample sizes of selected studies ranged from 729 [24] to 13,798 [29]. Just one article did not calculate the prevalence or incidence of multimorbidity [30]. The only longitudinal study showed the incidence of half year cumulative incidence and the incidence rate (IR) for multimorbidity [29]. Nine cross-sectional articles directly showed the number of participants

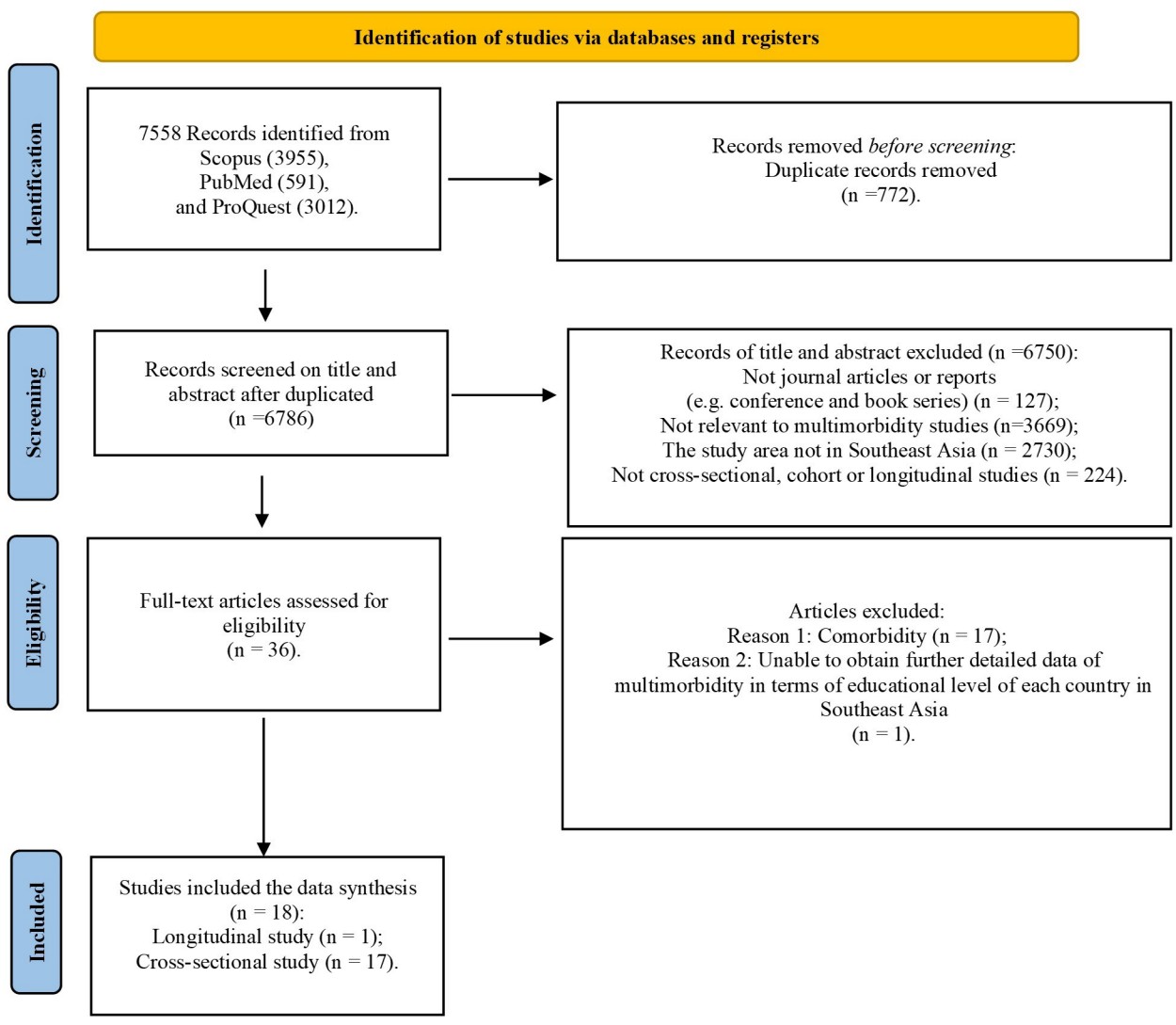

*From:* Page MJ, McKenzie JE, Bossuyt PM, Boutron I, Hoffmann TC, Mulrow CD, et al. The PRISMA 2020 statement: an updated guideline for reporting systematic reviews. BMJ 2021;372:n71. doi: 10.1136/bmj.n71

For more information, visit: http://www.prisma-statement.org/

**Fig 1. PRISMA flow diagram.** Description of article selection process.

experiencing multimorbidity at that educational attainment level and the total sample number to get the prevalence of multimorbidity at different educational levels [15,23–26,31–34]. Six studies included only participants aged 60 years and older [15,25,29,32,34,36].

On the NOS, six studies were scored Good level of quality [15,24,28,30,31,36] nine studies (including the only longitudinal study [29]) were at Fair level [14,16,23,25,26,29,33,34,37], and three studies were at Poor level [27,32,35] (Table 3).

**Table 3. The key characteristics of selected studies.**

| Study | Study area | | | | | Method | | | | | Results | Other |
|---|---|---|---|---|---|---|---|---|---|---|---|---|
| First author (year) | Country/ Location | Study population and study design | Sample size (N) | Men (%) | Mean or Medium age (years) (range) | Data collection | Definition of multimorbidity | Number of conditions | Educational level | Prevalence/Incidence of multimorbidity (95% CI)ᵃ | Prevalence/Incidence of multimorbidity in terms of educational level (%)ᵃ | Scores in NOS (Level) |
| **Cross-sectional studies** | | | | | | | | | | | | |
| Abdin (2020) [23] | Singapore | Singapore Mental Health Study (SMHS)-2010 and SMSH-2016; | SMHS-2010 = 6616 citizens; SMSH-2016 = 6126 citizens | SMHS-2010 = 48.5%; SMSH-2016 = 49.6% | Mean: N.A. ≥18 years | Fully structured diagnostic interview (mental disorders); Self-reported (physical disorders) | "Co-morbidity of mental and physical disorders." | 3 Mental disorders: "major depressive disorder, dysthymia and bipolar disorder; anxiety disorders, including generalised anxiety disorder and obsessive-compulsive disorder; and alcohol abuse and dependence"; 10 Physical disorders: "asthma, diabetes mellitus, hypertension and high blood pressure, chronic pain, cancer, cardiovascular disorders, ulcer and chronic inflamed bowel, thyroid disease, neurological condition, chronic lung disease." | Primary and below; Secondary; Diploma; Vocational; University | SMHS-2010 = 5.8% (5.3–6.4%); SMHS-2016 = 6.7% (6.1–7.3%); | SMHS-2010: Primary and below = 0.67% (0.48–0.90%); Secondary = 1.6% (1.3–1.9%); Diploma = 1.4% (1.1–1.7%); Vocational = 0.51% (0.36–0.71%); University = 1.2% (0.96–1.5%)  SMHS-2016: Primary and below = 1.1% (0.83–1.4%); Secondary = 1.7% (1.4–2.1%); Diploma = 1.7% (1.4–2.0%); Vocational = 1.2% (0.95–1.5%); University = 1.5% (1.2–1.9%) | 6 (Fair) |
| Afshar (2015) [16] | Laos; Malaysia; Myanmar; Philippines | WHO World Health Survey (WHS) | Laos = 4989; Malaysia = 6145; Myanmar = 6045; Philippines = 10083 | Laos = 49.3%; Malaysia = 50.4%; Myanmar = 48.9%; Philippines = 49.6% | Mean: N.A. (≥19years) | Self-reported | "The presence of two or more chronic diseases." | 6 conditions: "arthritis, angina or angina pectoris (a heart disease), asthma, depression, schizophrenia or psychosis, and diabetes." | < Primary; Primary school; Secondary Higher | (Standardised prevalence) Laos = 3.6% (3.1–4.1%); Malaysia = 5.6% (5.0–6.2%); Myanmar = 1.7% (1.4–2.0%); Philippines = 7.1% (6.6–7.7%); | N/A | 6 (Fair) |
| Anindya (2021) [24] | Indonesia | 2014/2015 Indonesian Family Life Survey (IFLS-5) | 13,798 adults | 49% | Mean: 58 years, Interquartile range (IQR): 54–65 years (≥40 years) | Self-reported; Medical examination | "The presence of two or more Chronic non-communicable diseases (NCDs)." | 14 NCDs: "high blood pressure, diabetes, asthma, heart attack/coronary heart disease, liver disease, stroke, cancer, arthritis/rheumatism, high cholesterol, (excluding malignancy), digestive disease, mental illness, memory related diseases." | No education; Primary; Junior high school; Senior high school; Tertiary | 20.84% (20.12–21.57%) | No education = 8.2% (7.8–8.7%); Primary = 5.1% (4.7–5.5%); Junior high school = 2.5% (2.2–2.7%); Senior high school = 4.0% (3.7–4.3%); Tertiary = 2.2% (2.0–2.5%) | 8 (Good) |
| Aye (2019) [25] | Myanmar | Community-based cross-sectional study | 4,859 participants | 37.9% | Mean: N.A. (60–106 years, 46.0% participants ≥70 years) | Face-to-face interview with a semi-structured paper questionnaire | "Two or more chronic conditions" | 14 chronic conditions: "high blood pressure, coronary heart disease or heart attack, heart failure, irregular heart-beat, chronic bronchitis or chronic obstructive airway disease, asthma, stroke, diabetes, arthritis or rheumatoid arthritis, osteoporosis, glaucoma, cataract, depression, and emotional or mental illness." | Diploma/ Graduate; Middle to High school; Below Middle school; Illiterate | 33.2% (31.9–34.5%) | Illiterate = 3.3% (2.8–3.8%); Below Middle school = 19.9% (18.5–20.8%); Middle to High school = 8.2% (7.4–8.9%); Diploma/graduate = 1.8% (1.4–2.1%) | 6 (Fair) |
| Ba (2019) [26] | Central Highlands Region (Tay Nguyen) of Vietnam | Cross-sectional study | 1680 people | 50.1% | Medium: 38.0 years, IQR: 30.5–43.0 years (≥15 years) | Self-reported | "The presence of two or more chronic conditions." | At least 8 chronic conditions: "cancer, heart and circulatory conditions, chronic joint problems, chronic pulmonary diseases, chronic kidney problems, chronic digestive problems, psychological illness, diabetes, and/or other chronic conditions." | Secondary or less; High school; University | 16.4% (14.6–18.2%) | Secondary or less = 9.5% (8.1–11.0%); High school = 2.4% (1.7–3.2%); University = 4.5% (3.6–5.6%) | 6 (Fair) |
| Chong (2012) [27] | Singapore | Singapore Mental Health Study (SMHS)-2010 | 6616 participants | 48.5% | 43.9 years (standard error (SE); ± 0.3 years) (≥18 years) | Face-to-face fully structured diagnostic interview (mental disorders); Self-reported (physical disorders) | "Co-morbidity of mental and physical disorders" | 5 Mental disorders: "major depressive disorder, bipolar disorder, generalised anxiety disorder, obsessive compulsive disorder, alcohol abuse and alcohol dependence"; 8 Physical disorders: "respiratory disorders (asthma, chronic lung disease), diabetes, hypertension and high blood pressure, chronic pain (arthritis or rheumatism, back problems including disk or spine, migraine headaches), cancer, neurological disorders (epilepsy, convulsion, Parkinson's disease), cardiovascular disorders (stroke or major paralysis, heart attack, coronary heart disease, angina, congestive heart failure or other heart disease), ulcer and chronic inflamed bowel (stomach ulcer, chronic inflamed bowel, enteritis, or colitis)." | Pre-primary; Primary; Secondary; Pre-U/Junior; College/ Diploma; Vocational; University | 6.1% (5.5–6.7%) | N/A | 6 (Poor) |

*(Continued)*

**Table 3.** (Continued)

| First author (year) | Study area Country/ Location | Study population and study design | Sample size (N) | Men (%) | Mean or Medium age (years) (range) | Method Data collection | Definition of multimorbidity | Number of conditions | Educational level | Results Prevalence/Incidence of multimorbidity (95% CI) | Prevalence/Incidence of multimorbidity in terms of educational level (%) | Other Scores in NOS (Level) |
|---|---|---|---|---|---|---|---|---|---|---|---|---|
| Ha (2015) [15] | Southern Vietnam | Community-based cross-sectional study | 2400 people | 34.8% | Mean: 72.6 years, standard deviation (SD): ±8.3 years (≥60 years) | Medical examination and chart review | "Having at least two of the conditions" | 6 broad groups of conditions: "cardiovascular (including hypertension), digestive system (including liver), respiratory (including chronic obstructive pulmonary disease and tuberculosis), arthritis (including osteoarthritis), genitourinary, and diabetes." | Illiterate; Literacy | 39.2% (39.5–43.8%) | Illiterate = 10.05% (8.9–11.3%); Literate = 31.4% (29.6–33.3%) | 7 (Good) |
| Hussain (2015) [28] | Indonesia | Indonesian Family Life Survey (IFLS-4) | 9438 Indonesia adults | 46.6% | Male medium: 52 years, IQR: 45–61 years (≥40 years); Female medium: 52 years, IQR: 45–62 years (≥40 years) | Active measurement or through self-report or both | "The presence of two or more chronic conditions in individual respondent" | 12 chronic conditions: "hypertension, diabetes, tuberculosis, asthma and other chronic lung diseases, cardiac diseases, liver diseases, stroke, cancer or malignancies, arthritis/ rheumatism, uric acid/gout, depression, vision and hearing abnormalities." | Elementary or less; High school; Graduate and above | (Age and sex standardised prevalence) 35.7% (34.8–36.7%) Men: 29.5% (28.1–30.8%); Women: 41.5% (40.1–42.8%). | N/A | 7 (Good) |
| Liew (2011) [30] | Indonesia | 2007 Indonesian Family Life Survey (IFLS4) | 3061 individuals | 51.2% | Male mean age: 54.01 years (40–93 years) Female mean age: 53.88 years (40–94 years) | Interview | "At least two chronic health conditions" | 10 chronic conditions: "hypertension, diabetes, tuberculosis, asthma, heart attack, liver disease, stroke, cancer, arthritis, gout." | Up to primary; Secondary; College and university | N.A. | N/A | 8 (Good) |
| Marthias (2021) [31] | Indonesia | Indonesian Family Life Survey (IFLS) conducted in Wave 4 2007 and Wave 5 2014 | Wave 4 (2007): 3678 respondents; Wave 5 (2014): 3678 respondents | 46.1% | Wave 4 (2007): Medium age: 58 years, IQR: 54–65 years (≥50 years); Wave 5 (2014): Medium age: 65 years, IQR: 60–72 years (≥50 years) | Self-reported and physical examination | "Two or more non-communicable diseases (NCDs)" | 10+4 NCDs: "Wave 4: hypertension, diabetes, asthma, heart attack/coronary heart diseases, liver disease, stroke, cancer, arthritis/rheumatism, high cholesterol, depression/mental illness (Wave 5 added: prostate diseases, kidney diseases (excluding malignancy), digestive diseases, memory-related diseases)." | No education; Primary; Junior high school; Senior high school; Tertiary | Wave 4 (2007) = 21.0% (19.6–22.6%) Wave 5 (2014) = 22.0% (20.6–23.6%) | Wave 4 (2007) No education = 11.6% (10.4–12.4%); Primary = 4.8% (4.2–5.6%); Junior high school = 1.9% (1.5–2.4%); Senior high school = 1.8% (1.4–2.3%); Tertiary = 1.2% (0.87–1.6%); Wave 5 (2014) No education = 10.3% (9.3–11.3%); Primary = 5.6% (4.9–6.4%); Junior high school = 2.7% (2.2–3.2%); Senior high school = 2.5% (2.0–3.0%); Tertiary = 1.6%. (1.2–2.0%) | 7 (Good) |
| Mwangi (2019) [32] | rural Northern Vietnam (FilaBavi) | FilaBavi Demographic Surveillance site in rural Vietnam | 2873 people | 41.6% | Mean: N.A. (≥60 years) | Self-reported | "Multiple diseases" | 8 non-communicable diseases (NCDs): "hypertension, diabetes, cancer, arthritis/ osteoarthritis, stroke, angina-pectoris, chronic bronchitis, cataract." | Illiterate; Read and write only; Primary/ Secondary; High school; Above high school | (Aggregated self-reported) 2 common chronic diseases (CCDs) = 9.2% (8.2–10.3%); ≥3 CCDs = 3.5% (2.8–4.1%) | ≥2 CCDs: Illiterate = 2.0% (1.5–2.6%); Read and write only = 4.9% (4.2–5.8%); Primary/secondary = 4.4% (3.7–5.2%); High school = 0.6% (0.35–0.95%); Above high school = 0.8% (0.51–1.2%) | 4 (Poor) |
| Pengpid (2017) [14] | Four Greater Mekong countries: Cambodia, Myanmar, Thailand, Vietnam | A cross-sectional survey | 6236 participants | 33.8% | Mean: 53.0 years, SD: ±16.8 years (18–94 years, 59.8% participants ≥50 years) | Interviewed with a structured questionnaire | "Two or more chronic conditions" | 21 chronic conditions: "asthma, chronic obstructive pulmonary disease, diabetes mellitus, hypertension, dyslipidaemia, coronary artery disease, cardiac failure, cardiac arrhythmias, stroke, arthritis, cancer, gout, Parkinson's disease, liver disease, kidney disease, thyroid disease, stomach and intestinal diseases, epilepsy, mental disorders." | Grade 0–5; Grade 6–11; Grade 12 or more | All: 72.6% (71.5–73.7%) two conditions: 28.6% (27.5–29.8%), three conditions: 22.4% (21.4–23.5%) and four or more conditions: 21.6% (20.6–22.6%) | N/A | 6 (Fair) |

*(Continued)*

**Table 3.** (Continued)

| Study | Study area | | | | | Method | | | | | Results | Other |
|---|---|---|---|---|---|---|---|---|---|---|---|---|
| First author (year) | Country/Location | Study population and study design | Sample size (N) | Men (%) | Mean or Medium age (years) (range) | Data collection | Definition of multimorbidity | Number of conditions | Educational level | Prevalence/Incidence of multimorbidity (95% CI) | Prevalence/Incidence of multimorbidity in terms of educational level (%) | Scores in NOS (Level) |
| Pengpid (2021) [33] | Thailand | A cross-sectional survey | 1409 attendees | 24.5% | Mean age in monk healer setting: 47.3 years. Mean age in health centre: 53.3 years; (≥19 years) | Self-reported | "Two or more chronic conditions" | 16 chronic conditions: "hypertension, heart attack or stroke, high blood cholesterol, diabetes, emphysema/asthma, sore joints, osteoporosis, cancer or malignancy, migraine headaches, ulcer, fatigue disorder, sleeping problem, common mental disorder: somatization, generalized anxiety disorder and major depression, substance use disorder." | Primary or less; Secondary; Post-secondary | All: 45.2% (42.6–47.9%) Monk healer: 23.0% (20.9–25.4%); Health centre: 22.1% (20.0–24.4%) | All: Primary or less = 26.2% (23.9–28.6%); Secondary = 9.9% (8.4–11.6%); Post-secondary = 9.1% (7.6–10.7%) Monk healer: Primary or less = 8.9% (7.5–10.6%); Secondary = 6.7% (5.5–8.2%); Post-secondary = 7.4% (6.1–8.9%) Health centre: Primary or less = 17.3% (15.3–19.3%); Secondary = 3.2% (2.3–4.3%); Post-secondary = 1.7% (1.1–2.5%) | 5 (Fair) |
| Picco (2016) [34] | Singapore | The Well-being of the Singapore Elderly (WiSE) study, a population-based, cross sectional study | 2565 citizen and permanent residents | 44.1% | Mean: N.A. (75% participants 60–74 years) | Self-reported | "Two or more of these chronic conditions being present in the one person at the same time" | 10 chronic conditions: "high blood pressure, heart trouble, stroke, transient ischemic attacks, diabetes, depression, arthritis or rheumatism, chronic obstructive pulmonary diseases, breathlessness or asthma, cancer." | No education; Some, but did not complete primary; Completed primary; Completed secondary; Completed tertiary | 51.5% (50.0–53.5%) | None = 12.4% (11.1–13.7%); Did not complete primary = 13.5% (12.2–13.7%); Primary = 13.3% (12.2–14.8%); Secondary = 8.3% (7.2–9.4%); Tertiary 4.3% (3.5–5.1%) | 6 (Fair) |
| Subramaniam (2014) [35] | Singapore | Singapore Mental Health Study (SMHS) a population-based, cross sectional study | 6616 participants | 49.9% (unweighted) 48.5% (weighted) | Mean: N.A. (≥18 years) | Self-reported | "Multiple chronic medical conditions (MCMC)" | 15 chronic conditions in 8 types: "respiratory disorders (asthma, chronic lung disease), diabetes, hypertension and high blood pressure, chronic pain (arthritis or rheumatism, back problems including disk or spine, migraine headaches), cancer, neurological disorders (epilepsy, convulsion, Parkinson's disease), cardiovascular disorders (stroke or major paralysis, heart attack, coronary heart disease, angina, congestive heart failure or other heart disease), ulcer and chronic inflamed bowel (stomach ulcer, chronic inflamed bowel, enteritis, or colitis)." | Primary and below; Secondary; Pre-U/Junior College/Diploma; Vocational; University | 15.0% (unweighted) 16.3% (weighted) (14.2–15.9%) | Primary and below = 5.4% (4.8–5.9%); Secondary = 6.7% (6.1–7.3%); Pre-U/Junior College/Diploma = 2.1% (1.7–2.4%); Vocational = 1.0% (0.73–1.2%); University = 2.0% (1.6–2.3%) | 5 (Poor) |
| Subramaniam (2017) [36] | Singapore | the Well-being of the Singapore Elderly (WiSE) study, a population-based, cross sectional study | 2565 respondents | 43.5% (unweighted) 44.0% (weighted) | Mean: N.A. (≥60 years) | Professional examination; self-reported | Comorbid Diabetes and Depression | 2 conditions: "diabetes and depression." | None; Some, but did not complete primary; Completed primary; Completed secondary; Completed tertiary | 2.8% (unweighted) 1.5% (weighted) (2.2–3.5%) | N/A | 8 (Good) |
| Tiparadol (2012) [37] | Thailand | Thai National Health Examination Survey III | 36,877 participants | 47.8% | Mean: N.A. (≥15 years) | Health examination | Coexistence of Diabetes and Hypertension. | 2 conditions: "diabetes and hypertension." | No formal education; Less than 6 years; Secondary; University | All: 3.2% (2.9–3.6%), | N/A | 6 (Fair) |

**Longitudinal studies**

(*Continued*)

**Table 3.** (Continued)

| Study | | Study area | | Method | | | | | | | | Results | | Other |
|---|---|---|---|---|---|---|---|---|---|---|---|---|---|---|
| First author (year) | | Country/ Location | Study population and study design | Sample size (N) | Men (%) | Mean or Medium age (years) (range) | Data collection | Definition of multimorbidity | Number of conditions | Educational level | Prevalence/Incidence of multimorbidity (95% CI)* | | Prevalence/Incidence of multimorbidity in terms of educational level (%)# | Scores in NOS (Level) |
| Hussin (2019) [29] | | Malaysia | Community-based longitudinal study; Follow up 0.5 year | 729 participants (349 without any chronic disease and 380 with one disease at baseline) | No disease = 51.3%; One disease = 49.2% | Mean: No disease at baseline = 68.3 years, SD: ±6.0 years (≥60 years); Mean: One disease at baseline = 69.4 years, SD: ±6.4 years (≥60 years) | Self-reported | "Co-occurrence of two or more diseases within a single individual" | 15 diseases: "hypertension, high cholesterol, diabetes, stroke, osteoarthritis, heart diseases, cataract/ glaucoma, renal failure, asthma, chronic obstructive pulmonary disease, tuberculosis, gout, hip fracture, thyroid disorders, cancer." | No schooling; 1–6 years; 7–11 years; 12 years and more | No disease at baseline half-of-year cumulative incidence for multimorbidity: 18.8% (incidence rates were 13.7 per 100 person-years); One disease at baseline Half-of-year cumulative incidence for multimorbidity: 40.9% (incidence rates were 34.2 per 100 person-years) | | N/A | 5 (Fair) |

N/A: Not available.

* If the type of studies could not extract or calculate the prevalence of multimorbidity, the cumulative incidence (CI, %) of multimorbidity could be extracted directly from the study or calculated. The formula was calculated by dividing the number of new multimorbidity cases in a given period by the number of subjects at risk in the population initially at risk of multimorbidity at the start of the study. If the CI for multimorbidity could neither be extracted nor calculated from the study, it should be denoted by N/A.

# If the type of studies could not extract or calculate the prevalence (%) of multimorbidity at different educational levels, it was treated in same way as if the type of studies could not extract or calculate the prevalence of multimorbidity.

### Definition and measure of multimorbidity

Twelve studies defined multimorbidity as two or more chronic conditions (Table 3) [14–16,24–26,28–31,33,34]. Two studies defined only multiple diseases but did not emphasize a specific number of diseases [32,35]. Four studies defined multimorbidity as the coexistence of two chronic conditions [23,27,36,37], but not the combined effects of additional conditions with reference to the index chronic condition, unlike the definition of co-morbidity. The number of chronic conditions measured were from two [23,36,37] to twenty-one [14] in these studies. Multimorbidity in twelve studies included psychological disorders in addition to physical disorders [14,16,23–28,31,33,34,36], and four studies' multimorbidity included tuberculosis (TB) (an infectious disease) [15,28–30].

To determine the outcome of the condition, seven studies used self-report to collect data [16,26,29,32–35] and four studies used a combination of medical or professional examination and self-reports [24,28,31,36]. But one study used only health examination [37].

### The prevalence of multimorbidity and multimorbidity in terms of education

The prevalence of multimorbidity ranged from 3.2% to 72.6% among over 15 years-old participants. Moreover, the prevalence ranged from 1.5% to 51.5% among older people (more than 60 years old) (Table 3).

The prevalence of multimorbidity was from 20.84% ($\geq$ 40 years old) [24] to 35.7% ($\geq$ 40 years old) [28] in Indonesia [24,28,30,31], and in all four studies [24,28,30,31] higher education related to higher odds of multimorbidity among over 40-years-old persons. The prevalence was from 1.5% [36] to 51.5% among people over 60 years old [34] and from 5.8% [23] to 45.2% [33] among people ($\geq$ 18 years old) in Singapore, and three [23,35,36] in all five studies [23,27,34–36] showed that education had no association with multimorbidity. The prevalence of multimorbidity was from 12.7% among older people ($\geq$ 60 years old) [32] to 39.2% among over 60-years-old people [15] in different parts of Vietnam [15,26,32], and two studies [15,26] suggested that education may reduce people's odds of developing multimorbidity but the age of the study population ($\geq$ 18 years vs. $\geq$ 60 years) for these two articles were different (Tables 3 and 4).

### Association between educational level and multimorbidity

There were three outcomes of these studies. First, higher education reduced odds of suffering from multimorbidity [14,15,26,27,30,33,34], second was that lower education reduced likelihood of having multimorbidity [24,35,28,30,31,33,37] and third was that educational attainment was not related to multimorbidity [16,23,28,29,32,33,35,36] (Table 4).

### Higher education reducing odds of multimorbidity

Seven cross-sectional studies [14,15,26,27,30,33,34] found education has been associated with reducing odds of multimorbidity. The study of Liew [30], found a higher level of education was beneficial for the health of over 40-year-old Indonesian women to fight against multimorbidity (all $p$-value $< 0.05$) after controlling for mobility problems, age, marital status, and smoking. Similar results were found in studies in other countries such as Vietnam [15,26]. For example, in accordance with the study of Ba *et al* in Central Highlands Region of Vietnam [26], after controlling for gender, age, education, and occupation, over 15-year-old participants with a high school education had significantly lower odds of suffering from multimorbidity compared to those with secondary or less school education (adjusted odd ratio (aOR) 0.8, 95%

**Table 4. The association between educational level and multimorbidity.**

| First author (year), Study area, Study design, Age range | The association between educational level and multimorbidity (value, (95% CI, p-value)) | | | | The main results of the relationship between educational level and multimorbidity | Adjusted factors |
|---|---|---|---|---|---|---|
| Abdin (2020) Singapore. Cross-sectional. ≥18 years [23] | **Educational level** | **SMHS-2010 & SMHS-2016** | **Year of survey interaction** | | **No association between educational level and multimorbidity.** After adjusting for all covariates, the likelihood of having comorbid mental and physical conditions over time was significantly lower in secondary school than those in university (aOR 0.5, 95% CI 0.3–0.9). | Age, sex, ethnicity, marital status, (education), and employment. |
| | Primary and below | OR 0.9 (0.6–1.5, p = 0.787) | (2016) OR 1.02 (0.5–2.2, p = 0.954) | | | |
| | Secondary | OR 1.2 (0.9–1.7, p = 0.266) | **(2016) OR 0.5 (0.3–0.9, p = 0.034)** | | | |
| | Diploma | OR 1.2 (0.9–1.7, p = 0.171) | (2016) OR 0.8 (0.5–1.6, p = 0.650) | | | |
| | Vocational | OR 1.3 (0.9–2.0, p = 0.178) | (2016) OR 1.2 (0.5–2.7, p = 0.631) | | | |
| | University (Reference) | OR 1.0 | (2010) OR 1.0 | | | |
| Afshar (2015) Laos, Malaysia, Myanmar, Philippines. Cross-sectional. ≥18years [16] | **Educational level** | **Laos** | **Malaysia** | **Myanmar** · **Philippines** | **No association between educational level and multimorbidity.** Educational levels were not associated with multimorbidity in these four countries after controlling the adjustment. (all p-value>0.05) | Age and gender. |
| | < Primary | OR 1.3 (p>0.05) | OR 1.1 (p>0.05) | OR 0.6 (p>0.05) · OR 1.6 (p>0.05) | | |
| | Primary school (Reference) | OR 1.0 | OR 1.0 | OR 1.0 · OR 1.0 | | |
| | Secondary | OR 0.5 (p>0.05) | OR 0.8 (p>0.05) | OR 1.5 (p>0.05) · OR 1.1 (p>0.05) | | |
| | Higher | OR 0.3 (p>0.05) | OR 0.8 (p>0.05) | OR 1.2 (p>0.05) · OR 0.7 (p>0.05) | | |
| Anindya (2021) Indonesia. Cross-sectional. ≥40 years [24] | **Educational level** | | | | **Higher educational level increasing odds of multimorbidity.** The prevalence of multimorbidity was greater in higher educated people. Participants with tertiary or higher education had more 1.60 times of odds (95% CI 1.32–1.93) to have multimorbidity compared with those who did not receive education after controlling for the confounders. | Age, gender, marital status, (education), residency, region, per capita expenditure (PCE) quartile and have or not health insurance. |
| | No education (Reference) | OR 1.00 | | | | |
| | Primary | **OR 1.22 (1.08–1.38, p = 0.002)** | | | | |
| | Junior high school | **OR 1.31 (1.11–1.54, p = 0.001)** | | | | |
| | Senior high school | **OR 1.27 (1.09–1.48, p = 0.002)** | | | | |
| | Tertiary | **OR 1.60 (1.32–1.93, p< 0.0001)** | | | | |
| Aye (2019) Myanmar. Cross-sectional. 60–106 years [25] | **Educational level** | | | | **Higher educational level increasing odds of multimorbidity.** The prevalence of multimorbidity was lower in the participants with less than middle school education compared to the reference group of those with a diploma (adjusted prevalence ratio (aPR) 0.50, 95% CI 0.25–0.99). | Residence, sex, (level of education), smoking, drinking, general health status and involved in social activities. |
| | Illiterate | PR 0.48 (0.22–1.05) | | | | |
| | Below Middle school | **PR 0.50 (0.25–0.99)** | | | | |
| | Middle to High school | PR 0.60 (0.29–1.27) | | | | |
| | Diploma/graduate (Reference) | PR 1.00 | | | | |
| Ba (2019) Central Highlands Region of Vietnam. Cross-sectional. ≥15 years [26] | **Educational level** | | | | **Higher educational level reducing odds of multimorbidity.** "After controlling for other variables, participants who received a high school education may have lower 0.8 times of odds (95% CI 0.59–0.98) of suffer from multimorbidity compared to those with secondary or less school education. | Sex, age, (education), and employment. |
| | Secondary or less (Reference) | OR 1.00 | | | | |
| | High school | **OR 0.8 (0.59–0.98)** | | | | |
| | University | OR 1.07 (0.72–1.60) | | | | |
| Chong (2012) Singapore. Cross-sectional. ≥18 years [27] | **Educational level** | **Any mental disorder only** | **Any physical disorder only** | **Comorbid mental-physical disorder** | **Higher educational level reducing odds of multimorbidity.** People with secondary education had more 2.1 times of odds (95% CI 1.2,3.8) of suffering from comorbid mental and physical disorders than those with university degree but the association between poor educational attainment and comorbid mental and physical conditions was not obvious. | N/A |
| | Pre-primary | - | OR 1.0 (0.6–1.6, p>0.05) | OR 0.6 (0.2–2.1, p>0.05) | | |
| | Primary | OR 0.8 (0.4–1.6, p>0.05) | OR 1.2 (0.8–1.7, p>0.05) | OR 0.8 (0.3–1.8, p>0.05) | | |
| | Secondary; Pre-U/Junior | OR 1.1 (0.6–1.8, p>0.05) | **OR 1.4 (1.1–1.9, p<0.05)** | **OR 2.1 (1.2–3.8, p<0.01)** | | |
| | College/Diploma | OR 0.8 (0.5–1.3, p>0.05) | OR 1.1 (0.8–1.5, p>0.05) | OR 1.4 (0.8–2.2, p>0.05) | | |
| | Vocational | OR 0.9 (0.5–1.6, p>0.05) | OR 1.2 (0.8–1.7, p>0.05) | OR 1.4 (0.7–2.8, p>0.05) | | |
| | University (Reference) | OR 1.0 | OR 1.0 | OR 1.0 | | |
| Ha (2015) Southern Vietnam. Cross-sectional. ≥60 years [15] | **Educational level** | | | | **Higher educational level reducing odds of multimorbidity.** People who were literate (aOR 0.68, 95% CI 0.54–0.85) would have lower likelihood of multimorbidity after controlling for the other variables. | Age, sex, marital status, (literacy), working status, residence, drinking, smoking, BMI, basic activities for daily activity and healthcare utilisation. |
| | Illiterate (Reference) | OR 1.00 | | | | |
| | Literacy | **OR 0.68 (0.54–0.85, p = 0.001)** | | | | |
| Hussain (2015) Indonesia, Cross-sectional, ≥40 years [28] | **Educational level** | **Men** | **Women** | | **Men: Higher educational level increasing odds of multimorbidity.** Higher educated men had higher odds of multimorbidity. **Women: No association between educational level and multimorbidity.** There was no association between education and multimorbidity in women. | Age, house location, ethnicity, (education), marital status, and per capita expenditure quintiles. |
| | Elementary or less (Reference) | OR 1.0 | OR 1.0 | | | |
| | High school | **OR 1.2 (1.0–1.5)** | OR 1.0 (0.9–1.2) | | | |
| | Graduate and above | **OR 1.5 (1.1–1.9)** | OR 1.2 (0.9–1.5) | | | |

*(Continued)*

**Table 4.** (*Continued*)

| First author (year), Study area, Study design, Age range | The association between educational level and multimorbidity (value, (95% CI, *p*-value)) | | | The main results of the relationship between educational level and multimorbidity | Adjusted factors |
|---|---|---|---|---|---|
| Liew (2011) Indonesia, Cross-sectional, Male: 40–93 years, Female: 40–94 years [30] | **Educational level:** Up to primary; Secondary; College and university | **Model 1** | **Model 2** (Building upon the Model one to add chronic health conditions, mobility problems, age, marital status, and smoking behaviour.) | **All: Higher educational level increasing odds of multimorbidity.** High educated people with have higher odds of suffering from at least two chronic health conditions. **Women: Higher educational level reducing odds of multimorbidity.** Higher level of education was beneficial for the health of females. Women (from around 2.2 to 1.6) with a college or university education suffered from a lower number of chronic illnesses than the male counterparts (from about 1.9 to 1.75). | Mobility problems, age, marital status, and smoking. |
| | Education (lower education level as reference) | Coefficients: 0.141 **p< 0.01** **OR 1.151** | Coefficients: 0.121 **p< 0.05** **OR 1.129** | | |
| | Female-Education (lower education level as reference) | Coefficients: −0.310 **p< 0.001** **OR 0.733** | Coefficients: −0.148 **p< 0.05** **OR 0.862** | | |
| Marthias (2021) Indonesia, Cross-sectional, ≥50 years [31] | **Educational level** | **Main results (IFLS-4 & IFLS-5)** | **Robustness check (IFLS-5)** | **Higher educational level increasing odds of multimorbidity.** **Mian results:** In the comparison with lower educational level, higher educated participants would be more likely to experience multimorbidity (aOR 1.50, 95% CI 1.12–2.02 for junior level; aOR 1.54, 95% CI 1.01–2.34 for tertiary level). **Robustness check:** Those people with higher education would be linked to a heavier burden of multimorbidity. | Gender, age, marital status, (education), ethnicity, insurance coverage, type of work and per capita household expenditure, residency and region. |
| | No education (Reference) | OR 1.00 | OR 1.00 | | |
| | Primary | OR 1.19 (0.98–1.44, *p* = 0.081) | **OR 1.35 (1.16–1.57, *p*<0.01)** | | |
| | Junior high school | **OR 1.50 (1.12–2.02, *p* = 0.007)** | **OR 1.66 (1.33–2.06, *p*<0.01)** | | |
| | Senior high school | OR 0.96 (0.71–1.29, *p* = 0.778) | OR 1.23 (0.99–1.53, *p*>0.05) | | |
| | Tertiary | **OR 1.54 (1.01–2.34, *p* = 0.043)** | **OR 1.77 (1.33–2.36, *p*<0.01)** | | |
| Mwangi (2019) rural Northern Vietnam (FilaBavi), Cross-sectional, ≥60 years [32] | **Educational level** | **having a common chronic disease (CCD)** | **having one CCD or more than one CCDs** | **No association between educational level and multimorbidity.** Old people with high school level had more odds of one common chronic disease (CCD) (OR 2.54, 95% CI 1.13–5.74, *p*-value <0.05). However, there was no association between education and having one CCD or more than one CCDs. | N/A |
| | Illiterate (Reference) | 1.00 | 1.00 | | |
| | Read and write only | OR 1.44 (0.89–2.34, *p* = 0.141) | OR 1.1 (0.452–2.681, *p* = 0.833) | | |
| | Primary/secondary | OR 1.38 (0.83–2.27, *p* = 0.213) | OR 0.904 (0.364–2.246, *p* = 0.828) | | |
| | High school | **OR 2.54 (1.13–5.74, *p* = 0.025)** | OR 2.103 (0.555–7.959, *p* = 0.274) | | |
| | Above high school | OR 1.93 (0.89–4.18, *p* = 0.096) | OR 0.607 (0.142–2.589, *p* = 0.5) | | |
| Pengpid (2017) Four Greater Mekong countries: Cambodia, Myanmar, Thailand, Vietnam, Cross-sectional, 18–94 years [14] | **Educational level** | **Two conditions *vs.* one condition** | **Three or more conditions *vs.* one condition** | **Higher educational level reducing odds of multimorbidity.** In comparison with those who had only one chronic condition, Lower educated people had more odds of having multimorbidity. | Gender, age, (education), income, region, quality of life and physical inactivity. |
| | Grade 0–5 (Reference) | OR 1.00 | OR 1.00 | | |
| | Grade 6–11 | **OR 0.78 (0.62–0.99, *p*<0.05)** | **OR 0.44 (0.36–0.54, *p*<0.001)** | | |
| | Grade 12 or more | **OR 0.59 (0.44–0.78, *p*<0.001)** | **OR 0.30 (0.23–0.39, *p*<0.001)** | | |
| Pengpid (2021) Thailand, Cross-sectional, ≥19 years [33] | **Educational level** | **Monk healer** | **Primary care** | **All** | **Monk healer: Higher educational level increasing odds of multimorbidity.** People with post-secondary education had more odds of multimorbidity in the monk healer setting (aOR 1.68, 95% CI 1.03, 2.76). **Primary care: Higher educational level reducing odds of multimorbidity.** Participants with secondary education had less odds of multimorbidity of multimorbidity (aOR 0.47, 95% CI 0.29–0.75) when comparing to primary or less education in the primary care setting. **All: No association between educational level and multimorbidity.** There was no association between education and multimorbidity in the combination of monk healer and primary care. | Gender, age, (education), employment, marital status, economic status, comorbidity, and health care setting. |
| | Primary or less (Reference) | OR 1.00 | OR 1.00 | OR 1.00 | | |
| | Secondary | OR 1.20 (0.74–1.95, *p*>0.05) | **OR 0.47 (0.29–0.75, *p*<0.01)** | OR 0.72 (0.52–1.00, *p*>0.05) | | |
| | Post-secondary | **OR 1.68 (1.03–2.76, *p*<0.05)** | OR 0.83 (0.41–1.67, *p*>0.05) | OR 1.27 (0.87–1.86, *p*>0.05) | | |
| Picco (2016) Singapore, Cross-sectional, Major: 60–74 years [34] | **Educational level** | | | **Higher educational level reducing odds of multimorbidity.** People who had secondary education would have less odds of suffering from multimorbidity (aOR 0.6, 95% CI 0.3–0.9, *p* = 0.047). | Age, sex, ethnicity, marital status, (education), and employment. |
| | No education (Reference) | OR 1.0 | | | | |
| | Some, but did not complete primary | OR 0.8 (0.5–1.3, *p* = 0.342) | | | | |
| | Completed primary | OR 0.7 (0.4–1.2, *p* = 0.156) | | | | |
| | Completed secondary | **OR 0.6 (0.3–0.9, *p* = 0.047)** | | | | |
| | Completed tertiary | OR 0.6 (0.3–1.2, *p* = 0.123) | | | | |

(*Continued*)

**Table 4.** (Continued)

| First author (year), Study area, Study design, Age range | The association between educational level and multimorbidity (value, (95% CI, *p*-value)) | | | The main results of the relationship between educational level and multimorbidity | Adjusted factors |
|---|---|---|---|---|---|
| Subramaniam (2014) Singapore, Cross-sectional, ≥18 years [35] | **Educational level** | | | **No association between educational level and multimorbidity.** Educational level was not associated with two or more chronic medical conditions (all the *p*>0.05). | N/A |
| | Primary and below | OR 1.0 (0.6–1.7, *p* = 0.93) | | | |
| | Secondary | OR 1.3 (0.9–2.0, *p* = 0.23) | | | |
| | Pre-U/Junior College/Diploma | OR 0.9 (0.6–1.4, *p* = 0.61) | | | |
| | Vocational | OR 1.0 (0.6–1.8, *p* = 0.98) | | | |
| | University (Reference) | OR 1.0 | | | |
| Subramaniam (2017) Singapore, Cross-sectional, ≥60 years [36] | **Educational level** | **Model 1** | **Model 2** (Having two additional adjusted factors compared with Model 1: Global Cognitive Score (COGSCORE) and World Health Organization Disability Assessment Schedule II (WHODAS II)) | **No association between educational level and multimorbidity.** There was not an association between education and comorbid depression and diabetes mellitus (DM) (all the *p*>0.05). | Model 1: Age, sex, ethnicity, marital status, (education), employment, obesity/ overweight, smoking, diabetes treatment, any other chronic condition. Model 2: Age, sex, ethnicity, marital status, (education), employment, obesity/ overweight, smoking, diabetes treatment, any other chronic condition COGSCORE and WHODAS II. |
| | None | OR 2.9 (0.3–30.2, *p* = 0.379) | OR 3.4 (0.6–18.7, *p* = 0.167) | | |
| | Some, but did not complete primary | OR 0.6 (0.1–5.4, *p* = 0.608) | OR 1.1 (0.2–6.1, *p* = 0.871) | | |
| | Completed primary | OR 0.8 (0.1–7.2, *p* = 0.806) | OR 1.3 (0.2–8.5, *p* = 0.795) | | |
| | Completed secondary | OR 0.7 (0.1–4.1, *p* = 0.708) | OR 0.8 (0.2–3.0, *p* = 0.749) | | |
| | Completed tertiary (Reference) | OR 1.00 | OR 1.00 | | |
| Tiptaradol (2012) Thailand, Cross-sectional, ≥15 years [37] | **Educational level** | **Compared to those suffering from either diabetes or hypertension alone** | | **Higher educational level increasing odds of multimorbidity.** People with education less than 6 years (aOR 1.83, 95% CI 1.03–3.38) had more odds of suffering from the coexistence of both conditions after controlling for potential confounding factors of sociodemographic variable. | Age, gender, residence, (education), region, BMI, and abdominal obesity (waist circumference ≥90 cm in male and ≥80 cm in female). |
| | No formal education (Reference) | OR 1.00 | | | |
| | Less than 6 years | **OR 1.83 (1.03, 3.38)** | | | |
| | Secondary | OR 0.96 (0.54, 1.72) | | | |
| | University | OR 1.06 (0.56, 2.01) | | | |
| Hussin (2019) Malaysia, Longitudinal, (0.5-year follow-up), ≥60 years [29] | **Educational level:** No schooling; 1–6; 7–11; 12 years and above | **No disease at baseline** | **One disease at baseline** | **No association between educational level and multimorbidity.** Without any disease at baseline showed that education was not related to multimorbidity incidence at follow-up (*p*>0.05). With one disease at baseline showed that education was not related to multimorbidity incidence at follow-up (*p*>0.05). | Without any disease at baseline: age, gender, (education), smoking, cognitive and lifestyle. With one disease at baseline: age, sex, (education), BMI, glucose, cognitive and dietary intake. |
| | 0–6 years | OR 1.296 (0.555–3.027, *p* = 0.549) | OR 0.584 (0.320–1.064, *p* = 0.079) | | |
| | 7 and above (Reference) | OR 1.000 | OR 1.000 | | |

Bolded font of number indicated a significant difference.

N/A: Not available.

Confidence interval (CI) 0.59–0.98). Furthermore, in the other study focused on the population older than 60 years in southern Vietnam [15], literate individuals (aOR 0.68, 95% CI 0.54–0.85) were associated with lower odds of multimorbidity in comparison with illiterate individuals after controlling for the related confounders (Table 4).

## Higher education increasing odds of multimorbidity

Seven cross-sectional studies found education has been associated with increasing odds of multimorbidity [24,25,28,30,31,33,37] and four of these studies [24,28,30,31] were located in Indonesia. Anindya *et al* [24] showed that higher educated people (more than 40-year-old) had more odds of suffering from multimorbidity after controlling for socio-demographic variables (all *p*-value > 0.05). Moreover, in another study in Indonesia [31], comparing lower

levels of education to high level of education, 50-year-old and older participants would be more likely to suffer from multimorbidity. Furthermore, another study in Thailand [37] showed, after adjusting for potential confounders for socio-demographic variables, those 15 years old and older with less than 6 years of education (aOR 1.83, 95% CI 1.03–3.38) were more likely to have multimorbidity compared to those with no formal education (Table 4).

### Education having no association with multimorbidity

There were eight studies (including the only longitudinal study) which showed no association among educational attainment and multimorbidity [16,23,28,29,32,33,35,36]. Three cross-sectional studies [23,35,36] were located in Singapore. For instance, one study [35] showed that the odds of having multiple chronic medical conditions were not associated with educational attainment, but this study did not control for relevant confounders. Furthermore, in other countries like Laos, Malaysia, Myanmar, and Philippines [16], the results showed that educational levels were not associated with multimorbidity in these four countries after controlling for gender and age (all $p$-value > 0.05) (Table 4).

## Discussion

This was the first study to systematically review and assess the available literature on the prevalence of multimorbidity and the relationship between education level and multimorbidity in Southeast Asia.

We identified a small number of relevant publications and found heterogeneity between these studies. For example, the estimation of sample size, the grouping of education levels, the age groups of study participants, and the inclusion and exclusion criteria varied considerably among the studies we included, making comparability difficult [18] that did not allow us to clarify the association between education level and multimorbidity using meta-analysis.

There were two obvious associations between education and multimorbidity in Singapore and Indonesia, respectively. Firstly, educational level was not associated with multimorbidity in Singapore, alternately higher levels of education were associated with higher odds of developing multimorbidity in Indonesia. The reasons for these results may be related to differences in the purpose and methods of these studies, such as different patterns of multimorbidity, classification of education levels, and the confounding factors, and may also be related to the databases used in these studies.

Most studies in Singapore [23,35,36] mentioned that education level was not associated with multimorbidity. According to the authors, this is likely to be related to the fact that these studies [23,27,34–36] from Singapore used only two databases, the Singapore Mental Health Study (SMHS) [23,27,35] and the Wellbeing of the Elderly in Singapore (WiSE) [34,36]. It may be that the association between education level and multimorbidity was not significant in these two databases [23,27,34–36]. In addition, it is worth noting that the results would be influenced by many factors, even when a common data source was used for the analysis, the definition of multimorbidity, adjustment factors, and differences in education level, may impact on the results. For example, the study of Chong et al [27] and the study of Subramaniam et al [35] used the SMHS-2010, but the classification of education level (the study of Subramaniam et al [35] combining the level of Pre-U/Junior College/Diploma together) and the pattern of multimorbidity differed (the pattern of multimorbidity including mental disorders in the study of Chong et al [27]), thus the results of the connection of multimorbidity and education of these two studies were different. Similarly, the study of Picco et al [34] and the study of Subramaniam et al [36] used WiSE, but their patterns of multimorbidity and adjustment

factors were very different, leading to distinct results of the association between educational level and multimorbidity.

While all four Indonesia studies found [24,28,30,31] that higher education levels were associated with higher odds of developing multimorbidity, all of them used the Indonesian Family Life Survey (IFLS). It was likely that the association between higher education level and higher odds of developing multimorbidity was evident in the IFLS database. However, there were still many differences between the results of these studies. The reasons for this include factors such as the purpose of the study, the selection of the population, and the method of analysis affected the association between multimorbidity and education level. Also relevant was that the ILFS is a longitudinal study collecting data every 4–5 years. The various studies using ILFS data conducted cross sectional analysis on various waves of this study, which may also help explain the variation. A typical example was the study by Marthias *et al* [31]. This study consisted of two cross-sectional sub-studies whose purpose and methods were consistent. However, the difference in the pattern of multimorbidity and participants between these two sub-studies made the results of the association between multimorbidity and education level different [31].

High levels of education have been reported to reduce the odds of developing multimorbidity in western countries such as the United States and Canada [8]. However, in some studies of LMICs, the relationship between education level and multimorbidity was more complex than in developed countries. For example, in some studies from Bangladesh [38], India [39–41] and China [42,43], there was a positive [38,39,42], negative [40,43] or no association [41] between high education level and higher odds of multimorbidity. This was the same the results of our study.

The improvement of educational level was recognized as reducing risk of multimorbidity [10,44]. The potential reason would be that as the level of education increased, people's SES would also increase, making them more aware of healthy living and gaining health literacy and decreasing the risk [8,10,44]. In developed countries such as the United States and Canada, people with low SES were more likely to suffer from multimorbidity due to their lack of health knowledge, higher stress levels, and inability to afford healthy and adequate diet, which made them less healthy [8,10]. At the same time, their inability to afford the high cost of medical care after suffering from multiple diseases caused their health level to further worsen, which would be a vicious cycle [8,10]. The same problem was faced in developing countries. However, few studies have investigated multimorbidity in depth in developing countries, and the current focus in developing countries was still on single diseases [9,18]. This was because although aging has been also increasing in LMICs, the proportion of the elderly population is relatively low compared to that in developed areas, and aging is positively correlated with the incidence of multimorbidity [2,3,5]. These have led to under-research of multimorbidity in LMICs and a great variation from study to study, resulting in a diversity of results between multimorbidity and factors such as education level.

Several studies have shown that higher levels of education increase the chances of developing multimorbidity, and the reasons for this phenomenon include the following. Firstly, although people of lower educational level in LMICs were more likely to consume a poor diet quality–low in fruit and vegetables and high in red met and processed food–higher educated people tended to favour sedentary lifestyle habits, thus increasing the odds of developing multimorbidity [38]. In addition, people with higher levels of education had better access to medical care and better health knowledge, especially in developing countries. Since people with low SES in developing countries had more difficulty accessing effective medical care with relevant health knowledge compared to their counterpart in high-income countries [24,38]. These highly educated people were more likely to receive a diagnosis of their condition and to be reported as having chronic conditions [24]. As for the results of no association between

education and multimorbidity, it may be due to information bias in the study itself, selection bias, and confounding factors that have an impact on the results [35,41].

Most studies in this review did not describe the criteria for selecting chronic conditions [18,22]. Most common was selection of common or high prevalence diseases for inclusion in the study. For example, some studies included infectious diseases such as tuberculosis (TB) [15,28–30] and mental diseases [14,16,23–28,31,33,34,36], and the definition of multimorbidity in four [23,27,36,37] studies was two comorbid conditions. Because the number and type of chronic diseases determined the estimates of multimorbidity, prevalence rates and associations with education differed [18,22]. What is more, differences in study location, size and characteristics of the study population, data collection methods, and educational attainment classification may lead to selection bias and thus to different results [18,22]. Different data collection methods could also result in bias, and the recall bias of self-report could be more pronounced compared to medical examination for the identification of multimorbidity. In addition, different confounders had a significant impact on the results. For example, education level can be confounded by other socio-demographic factors including age, gender, and income [18,19,22]. Generally, the older age of the studied population implied a higher prevalence of multimorbidity, as we found in this paper [2,3,5,45]. As for the classification of educational attainment, the same data using different classifications of educational attainment could also lead to differences in the results that emerged, such as in the two studies [23,27] in Singapore.

## Strength and limitation

The main strength of this paper included the systematic listing of education level separately in relation to multimorbidity, rather than including it in the SES, which provided a more precise understanding of the association between education level and multimorbidity.

The results of most studies [16,32,33,35] obtained with self-report data showed no association between education level and multimorbidity, and the sample size and age range of these studies varied widely, suggesting that such results may be related to self-reported having recall bias. The studies of multimorbidity tended to favour objective test of multimorbidity such as physical examination, or a combination of physical examination (major) and self-report (minor), because this could reduce the bias in the outcomes. In addition, in studies with participants older than 40 years, most of the results [24,25,28,30,31] showed that high education level was associated with high odds of having multimorbidity. This may be due to the same set of datasets used in number of these studies [24,28,30,31]. We did not find a pattern in the effect of sample size on the results. However, in general, the larger the sample size, the more accurate the results should be, controlling for other variables such as age and data collection [8,19,22].

The differences in inclusion and exclusion criteria for chronic diseases, sample size estimates, differences in study areas, different age groups of study participants, differences in data collection methods, distinction in educational level groupings, differences in confounding factors may all lead to large biases across studies and, to some extent, explain the large differences in observed outcomes. The inherent bias in the estimates of the original studies prevented the assessment of the quantification of prevalence of multimorbidity and the estimation of the association between education level and multimorbidity. The similar heterogeneity was seen in the systematic review of studies on the prevalence of multimorbidity in South Asia [18], with different methodologies and research settings contributing to this phenomenon.

Additionally, there are seventeen cross-sectional studies but only one longitudinal study. The lack of longitudinal studies made it impossible to definitively state whether there was a causal relationship between education level and multimorbidity [22,45].

Another limitation was that it was difficult to ensure that all relevant literature was included. Since multimorbidity was not well indexed in literature databases and was often used interchangeably with the term "comorbidity" [18,22]. Although the term "comorbidity" was also searched separately in the database after the initial search to compensate for this search omission, inadvertent omissions could not be excluded [22]. In addition, other keywords such as "education level" and "Southeast Asia" may have other names that we did not mention, and some literature may have been omitted. An inherent limitation of any systematic review was the limitation of the search period, which in our case was from January 1, 1990, to June 15, 2021, implying the exclusion of new studies after the end date, which may have led to the omission of more recent studies [18]. We also restricted the search to English publications, resulting in relevant articles in other languages that we could not retrieve [18,22]. Furthermore, another major limitation was the large amount of statistical and methodological heterogeneity, causing the inability to combine studies to obtain overall estimates of prevalence in multimorbidity and the overall estimates of association of education level with multimorbidity [18].

## Implications for policy, practice, and future research

The Association between multimorbidity and education remains an underappreciated area of research in Southeast Asia. As mentioned above, the small number of relevant studies included in this article and the large differences of statistics and methodology between individual studies. The results of the available global systematic review [8] on SES and multimorbidity suggest that the higher the level of education is, the lower the odds of developing multimorbidity people have. Although this result is inconsistent with the association between multimorbidity and education in Southeast Asia derived in this paper, the salient heterogeneity of this study imposes limitations on the overall estimation of the connection between education level and multimorbidity to calculate a uniform association. Therefore, in the next studies in Southeast Asia, the panel of chronic conditions should be prepared with a standardized definition of each disease and a uniform operational definition of multimorbidity, which could reduce the selection bias of chronic conditions and lead to a more reliable and comparable estimate of multimorbidity [18,22]. In addition, there should be uniform and objective criteria for the classification of education level, such as no education; elementary school; middle school, junior high school, high school, university, or higher, and reducing self-reported data collection methods and strengthening objective tests, such as physical examinations, would also bring accurate results.

## Conclusion

This study is a comprehensive mapping of research related to the association between education and multimorbidity in the Southeast Asian region and reveals the neglect and lack of studies on the connection between multimorbidity and educational level in this region. The heterogeneity of the findings did not allow us to reach a definitive conclusion about the association between educational level and multimorbidity. The prevalence of multimorbidity ranged widely in this study, and the associations between educational attainment and multimorbidity were inconsistent. Reasons for this result include the different national contexts and the lack of relevant studies on the relationship between educational status and multimorbidity in Southeast Asian countries. This study may indicate the need to reduce the use of subjective data collection methods, such as self-report, to improve credibility and accuracy when studying the relationship between multimorbidity and education. Finally, there were several different connections between educational attainment and multimorbidity in Southeast Asia in this paper,

however, it is predicted that a sound and completed educational system could help people to raise health awareness and thus effectively prevent multimorbidity.

## Supporting information

**S1 Table. Search query of databases.**
(DOCX)

**S2 Table. NOS checklist for selected studies (cross-sectional study).**
(DOCX)

**S3 Table. PRISMA 2009 checklist.**
(DOC)

**S1 File. The PROSPERO-registered number-CRD42021259311.**
(PDF)

**S2 File. The Newcastle-Ottawa Scale (NOS) checklist.**
(PDF)

## Acknowledgments

Authors thank Dr. Nasser Bagheri for his help with the content and structure of the article.

## Author Contributions

**Conceptualization:** Xiyu Feng.

**Data curation:** Xiyu Feng, Matthew Kelly, Haribondhu Sarma.

**Formal analysis:** Xiyu Feng, Matthew Kelly, Haribondhu Sarma.

**Methodology:** Matthew Kelly.

**Writing – original draft:** Xiyu Feng.

**Writing – review & editing:** Xiyu Feng, Matthew Kelly, Haribondhu Sarma.

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
