## [Decision Letter · Decision Letter 0]

21 Oct 2021

PONE-D-21-29541The association between educational level and multimorbidity among adult in Southeast Asia: Systematic reviewPLOS ONE

Dear Dr. Kelly,

Thank you for submitting your manuscript to PLOS ONE. After careful consideration, we feel that it has merit but does not fully meet PLOS ONE’s publication criteria as it currently stands. Therefore, we invite you to submit a revised version of the manuscript that addresses the points raised during the review process.

Major revisions are needed in the present form. See the Reviewers' comments carefully and respond them appropriately.

We look forward to receiving your revised manuscript.

Kind regards,

Masaki Mogi

Academic Editor

PLOS ONE

Journal Requirements:

2. Please provide a table reporting in detail the results of your quality assessment, showing how each included study scored on every item of the NOS quality assessment tool.

Reviewers' comments:

Reviewer's Responses to Questions

**Comments to the Author**

1. Is the manuscript technically sound, and do the data support the conclusions?

Reviewer #1: Yes

Reviewer #2: Yes

2. Has the statistical analysis been performed appropriately and rigorously? 

Reviewer #1: N/A

Reviewer #2: Yes

3. Have the authors made all data underlying the findings in their manuscript fully available?

Reviewer #1: Yes

Reviewer #2: Yes

4. Is the manuscript presented in an intelligible fashion and written in standard English?

Reviewer #1: Yes

Reviewer #2: No

5. Review Comments to the Author

Reviewer #1: The paper was well constructed. The authors present clear definitions of exposures and outcomes, and pointed out the gaps that lead them to the study. Also, the aims have been clearly described. The methods used, as well as the results, meet the aim initially proposed. The authors clearly discuss the results and limitations of the study.

The manuscript is presented in standard English, although authors may consider proofreading before the next submission.

Reviewer #2: Thank you so much for submitting this interesting article. However, there are a few flaws in the manuscript that require attention.

Overall:

Please rewrite the entire manuscript in a more concise style.

-Introduction:

1. Line 69 and 71: References for higher mortality rate and not receiving cost-effective treatment were missing

2. Line 69, line 78: The sentence is way too long for readers to follow

-Method:

1. Please revise the entire article, especially the method session, in a concise fashion.

2. Line 174 to 204: the details could be put into a table or list.

3. Line 221 to 235, Line 247 to 251: Please rewrite it concisely.

-Result

1. The resolution of graph 1 could be improved

2. Please do not repeat all the information listed in the table; presenting the most critical findings could make this article clearer.

-Discussion

1. Line 396: The logic development of this argument is unclear. How does the different grouping method eliminate the potential association?

2. Line 411: How do the different times of data collection impact the association? Please elaborate more on this.

3. Line 426: The reference you cited here is a meta-analysis on 24 cross-sectional studies; this study design is unlikely to provide proof for causal correlation. Please check.

6. PLOS authors have the option to publish the peer review history of their article (what does this mean?). If published, this will include your full peer review and any attached files.

Reviewer #1: **Yes: **Jonas Eduardo Monteiro dos Santos

Reviewer #2: No

---

## [Author Response · Author response to Decision Letter 0]

10 Nov 2021

We have addressed the editors specific comments here. Responses to the reviewers are contained in the attached file of that name. 

We have edited out manuscript to meet the journal style. 

2. Please provide a table reporting in detail the results of your quality assessment, showing how each included study scored on every item of the NOS quality assessment tool.

We have now included this information as S2 Table

We have now added captions to the end of the manuscript for the Supporting Information file.s

---

## [Decision Letter · Decision Letter 1]

17 Nov 2021

PONE-D-21-29541R1The association between educational level and multimorbidity among adult in Southeast Asia: Systematic reviewPLOS ONE

Dear Dr. Kelly,

Thank you for submitting your manuscript to PLOS ONE. After careful consideration, we feel that it has merit but does not fully meet PLOS ONE’s publication criteria as it currently stands. Therefore, we invite you to submit a revised version of the manuscript that addresses the points raised during the review process.

Minor revisions are needed in the present form. See the Reviewers' comments carefully and respond them appropriately.

We look forward to receiving your revised manuscript.

Kind regards,

Masaki Mogi

Academic Editor

PLOS ONE

Journal Requirements:

Reviewers' comments:

Reviewer's Responses to Questions

**Comments to the Author**

1. If the authors have adequately addressed your comments raised in a previous round of review and you feel that this manuscript is now acceptable for publication, you may indicate that here to bypass the “Comments to the Author” section, enter your conflict of interest statement in the “Confidential to Editor” section, and submit your "Accept" recommendation.

Reviewer #1: (No Response)

2. Is the manuscript technically sound, and do the data support the conclusions?

Reviewer #1: Yes

3. Has the statistical analysis been performed appropriately and rigorously? 

Reviewer #1: N/A

4. Have the authors made all data underlying the findings in their manuscript fully available?

Reviewer #1: Yes

5. Is the manuscript presented in an intelligible fashion and written in standard English?

Reviewer #1: No

6. Review Comments to the Author

Reviewer #1: Dear authors, congratulations on the review work. I have made some suggestions for changing the text and hope it will be clearer to your future readers. However, I suggest a careful grammar and tenses revision.

7. PLOS authors have the option to publish the peer review history of their article (what does this mean?). If published, this will include your full peer review and any attached files.

Reviewer #1: **Yes: **Jonas Eduardo Monteiro dos Santos

---

## [Author Response · Author response to Decision Letter 1]

24 Nov 2021

We have responded to all comments in the attached 'response to reviewers' document.

---

## [Decision Letter · Decision Letter 2]

6 Dec 2021

The association between educational level and multimorbidity among adults in Southeast Asia: Systematic review

PONE-D-21-29541R2

Dear Dr. Kelly,

We’re pleased to inform you that your manuscript has been judged scientifically suitable for publication and will be formally accepted for publication once it meets all outstanding technical requirements.

Kind regards,

Masaki Mogi

Academic Editor

PLOS ONE

Additional Editor Comments (optional):

No further comment.

Reviewers' comments:

Reviewer's Responses to Questions

**Comments to the Author**

1. If the authors have adequately addressed your comments raised in a previous round of review and you feel that this manuscript is now acceptable for publication, you may indicate that here to bypass the “Comments to the Author” section, enter your conflict of interest statement in the “Confidential to Editor” section, and submit your "Accept" recommendation.

Reviewer #1: All comments have been addressed

2. Is the manuscript technically sound, and do the data support the conclusions?

Reviewer #1: Yes

3. Has the statistical analysis been performed appropriately and rigorously? 

Reviewer #1: Yes

4. Have the authors made all data underlying the findings in their manuscript fully available?

Reviewer #1: Yes

5. Is the manuscript presented in an intelligible fashion and written in standard English?

Reviewer #1: Yes

6. Review Comments to the Author

Reviewer #1: (No Response)

7. PLOS authors have the option to publish the peer review history of their article (what does this mean?). If published, this will include your full peer review and any attached files.

Reviewer #1: **Yes: **Jonas Eduardo Monteiro dos Santos

---

## [Editor Report · Acceptance letter]

10 Dec 2021

PONE-D-21-29541R2 

The association between educational level and multimorbidity among adult in Southeast Asia: Systematic review 

Dear Dr. Kelly:

I'm pleased to inform you that your manuscript has been deemed suitable for publication in PLOS ONE. Congratulations! Your manuscript is now with our production department. 

Kind regards, 

on behalf of

Dr. Masaki Mogi 

Academic Editor

PLOS ONE